# A Novel Physical Mobility Task to Assess Freezers in Parkinson’s Disease

**DOI:** 10.3390/healthcare11030409

**Published:** 2023-01-31

**Authors:** Lígia Reis Nóbrega, Eduardo Rocon, Adriano Alves Pereira, Adriano de Oliveira Andrade

**Affiliations:** 1Faculty of Electrical Engineering, Federal University of Uberlândia, Uberlândia 38400-902, MG, Brazil; 2Centre for Automation and Robotics (CAR), Spanish National Research Council and Higher Technical School of Industrial Engineering (CSIC-UPM), 28500 Madrid, Spain

**Keywords:** Parkinson’s disease, freezing of gait, mobility task, dual task, inertial sensors

## Abstract

Freezing of gait (FOG), one of the most disabling features of Parkinson’s disease (PD), is a brief episodic absence or marked reduction in stride progression despite the intention to walk. Progressively more people who experience FOG restrict their walking and reduce their level of physical activity. The purpose of this study is to develop and validate a physical mobility task that induces freezing of gait in a controlled environment, employing known triggers of FOG episodes according to the literature. To validate the physical mobility tasks, we recruited 10 volunteers that suffered PD-associated freezing (60.6 ± 7.29 years-old) with new FOG-Q ranging from 12 to 26. The validation of the proposed method was carried out using inertial sensors and video recordings. All subjects were assessed during the OFF and ON medication states. The total number of FOG occurrences during data collection was 144. The proposed tasks were able to trigger 120 FOG episodes, while the TUG test caused 24. The Inertial Measurement Unit (IMU) with accelerometer and gyroscope could not only detect FOG episodes but also allowed us to visualize the three types of FOG: akinesia, festination and trembling in place.

## 1. Introduction

Freezing of gait (FOG) is one of the most disabling features of Parkinson’s disease (PD) [1]. FOG motor disorder represents an intermittent failure to initiate or maintain locomotion [2]. FOG was defined in 2010, as a “brief episodic absence or marked reduction in stride progression despite the intention to walk” [3].

According to Bartels et al. [4], the prevalence of freezers, which are people diagnosed with PD in whom the FOG symptom manifests, ranges from 7% in the early stages to 60% in the severe stages. Rawson et al. [5] have stated that between 20% and 60% of people living with PD will eventually experience this symptom, and Saad et al. [6] considered that over half of patients with PD could develop FOG in the course of the disease. Morris et al. [7] have reported that progressively more people who experience FOG restrict their walking and reduce their level of physical activity to avoid triggering the motor disorder.

The relationship between akinesia and FOG remains unclear [4] as FOG is not necessarily a completely frozen akinetic posture [8]. Akinesia is the sudden inability to initiate any movement [9], and there is a type of FOG related to this behavior. However, different disturbances may be seen during the gait of freezers [8].

In 1995, Thompson and Marseden [8,10,11,12] defined three types of FOG, (I) freezing, when the feet seems like they are glued to the ground, as a complete or partial akinesia; (II) festination, when the normal gait rhythm changes to a shuffling gait, with faster and shorter steps; and (III) trembling in place, when the lower extremities may show signs of shaking while glued to the ground.

FOG episodes last more than one second and can be triggered by postural transitions [1]. An FOG trigger can be a movement, e.g., a turn while walking (turning hesitation), the beginning of walking (starting hesitation), and just before reaching a destination (destination hesitation); an environmental constraint, e.g., doors, narrow passages, public transportation, and small or messy spaces; a dual task, either motor dual task or motor-cognitive dual task; or a negative emotion, such as anxiety, stress, anger, fear, or distress [7,13,14,15].

FOG is still a poorly understood phenomenon [8] and its pathophysiology is still not clear enough [4]. Clinical evaluation of FOG has been included in the Movement Disorder Society Unified Parkinson’s Disease Rating Scale (MDS-UPDRS), the most widely used clinical scale for assessing PD [16]. In item 2.13 of Part II (MDS-UPDRS)—the motor experiences of daily living (M-EDL)—the evaluator asks the patient if, during a normal day in the past week, a sudden block or stop, as if his/her foot were glued to the floor, occurred while walking. The answer is a score from 0 (“No”) to 4 (“Due to FOG episodes, I need help to walk”). In item 3.11 of Part III (MDS-UPDRS), a motor examination—gait assessment—is performed while the patient walks away from and then towards the examiner, allowing both sides of the body to be observed simultaneously. The patient should walk at least 10 m (30 feet), then turn around and return to the evaluator. The evaluator looks for hesitations at the beginning and hesitation in movements, especially when turning around and reaching the end of the task. The evaluator scores the patient’s gait from 0 (no freezing) to 4 (the patient freezes several times while walking straight).

It is important to highlight that Part II of MDS-UPDRS has 13 items and Part III has 18 items in total, therefore, there are two specific items for FOG assessment out of 31. The MDS-UPDRS has the advantage of being available to most clinicians; however, it requires experience and may not reveal FOG even for cases confirmed by the individual’s medical history [8]. The assessment of FOG in the MDS-UPDRS is not an accurate representation of daily life and, as a consequence, FOG is often difficult to observe during a clinical visit [2].

A golden standard for the detection and assessment of the FOG phenomenon is currently unavailable [3,5,6,17]. The importance of this work is due to the crucial need for an accurate method of detecting FOG and rating its severity and impact for the determination of the appropriate treatment and follow up [5,6]. Furthermore, determining methods to assess physical mobility in PD could prevent falls, reduce or overcome FOG episodes, and increase the quality of life of freezers [5,6].

The need to assess physical mobility in clinical practice was raised by Podsiadlo and Richardson [18] in 1991, when the Timed-Up and Go (TUG) test, using a time score in seconds, was proposed to assess the risk of fall in elderly populations. Similarly, our hypothesis is that it is possible to assess FOG episodes in freezers with a simple and short test, using limited space.

Laboratory testing of gait motor abilities on sway platforms or treadmills is impractical in most clinical settings, due to their length and complexity; furthermore, it is not profitable to use them with frail patients [18]. The challenge to cause FOG in a controlled environment without the use of complex technologies and while respecting the fragility and limitation of the PD patient came about because of its convenience.

The objective of the current study is to develop a simple physical mobility task that induces FOG in a controlled environment using known triggers of FOG episodes as described in the literature in order to evaluate freezers in PD. Using Inertial Measurement Unit (IMU) with accelerometer and gyroscope and video recordings, the proposed method was validated.

## 2. Materials and Methods

### 2.1. Literature Review

The current study includes a review of the literature on the assessment of FOG in Parkinson’s disease. Our review was conducted to bring together the various methods used in related studies to cause FOG. The main databases used in our review were IEEE, Pubmed, Lilacs, and Medline. The papers considered to compose this study are listed in Table 1.

The following combinations of terms were used to maximize the scope and type of material referred in the search: (I) Parkinson (and) Freezing (and) Trigger (and) (clinic (or) clinical) (and) (pattern (or) standard); (II) Parkinson (and) Freezing (and) (clinic (or) clinical) (and) (behavior (or) conduct); (III) Parkinson (and) Freezing (and) Trigger (and) hypothesis; and (IV) Parkinson (and) Freezing (and) (different (or) distinct) (and) (clinic (or) clinical) (and) Cues. The search was carried in English.

#### Result of the Literature Review

Figure 1 depicts a flow diagram of the literature review. In Phase 1, the first column shows the total number of articles for each search term (i.e., I, II, III, and IV), the second column is the number of articles after excluding duplicates, and the third column is the number of articles left after reading the title and abstract. A total of 155 papers were fully read, and 50 studies left to compose the literature review. Tasks that could trigger FOG in a controlled environment are presented in 18 papers, from which all the motor triggers mentioned were included in the present study.

FOG is usually triggered by postural transitions [1], the triggers considered to be a possible manner of causing FOG, according to the literature review, are to sit and stand up from a chair, walk through a doorway, reach a destination, turn 180° and 360°, and carry out a dual task. The triggers considered in each study included in the literature review are shown in Table 1.

Table 1 shows 17 studies, it does not include the study of Saad et al. [6] because volunteer freezers were not recruited. Instead, their experiment used healthy researchers that simulated FOG during walking. The following paragraphs describe the studies included in the literature review.

Velik et al. [18] and Alvarez et al. [19] aimed to continuously monitor a patient in daily life instead of triggering FOG in a controlled environment, and Muralidharan et al. [25], Shine et al. [28], Killane et al. [21], and Waechter et al. [22] used virtual reality (VR) to induce FOG episodes. The VR experiments that explain the effect of sensory and cognitive processes on FOG are usually setups in which the patient navigates through a series of doorways while simultaneously responding to a cognitive task [21,22,24,25].

The virtual reality results are generally shown in motor arrests, defined as an instance where the step latency is twice the normal latency, this measure has a good correlation with the amount of FOG episodes during the classic TUG test [6,25]. However, there is a real need to compare the behavior obtained by VR models with actual walking tasks. VR has proven to be a reliable method of eliciting FOG episodes in a controlled clinical test environment [14,21,22,25]. On the other hand, VR is an expensive technology that requires previously trained staff that can compare results based on the behavior presented during the VR tasks, which simulate the effect of locomotion [25].

The literature review showed some studies [19,26] that were carried out in more spacious and wider environments with the aim of capturing and recording daily life motion and of detecting FOG. In Velik et al. [19], the researchers carried out an experiment where the subject executed motor dual tasks by going to different rooms in a house carrying objects, such as carrying clothes hangs to the laundry. The objective of the experiment was to quantify how sensory cues affect the duration of FOG episodes. To allow cueing, subjects wore a backpack with a small and lightweight laptop, which was remote controlled from another computer via Wi-Fi [19]. FOG detection in Velik et al. [19] was made by an assistant experienced in the recognition of FOG episodes. This person would observe the volunteer performing the course and trigger a cue (auditory, visual, or vibratory) always two seconds after a FOG episode occurred. To detect FOG episodes, Alvarez et al. [26] used a recurrent neural network (RNN). They extracted information from the trajectories of a 360-degree panoramic camera (Zenith), an RGB-D camera (Kinect), WSN sensors, inertial sensors (accelerometer, gyroscope, and magnetometer), and binary sensors placed on doors and drawers to detect when they are opened or closed [26].

To assess dual task abilities in patients with early-stage PD, Zirek et al. [29] applied the TUG test under single and dual task conditions. Their findings show that tasks that increase the demand for complex attention are more sensitive to showing impaired dual task ability. However, the referred study did not focus on freezers, in fact, one of the inclusion criteria was having a score of 3 or less in the New FOG-Q. Therefore, no FOG analysis was considered.

Beck et al. [15] aimed to explore how the interaction between cognitive and sensorial perception influences on FOG, the results advise that although increasing demand on attention does substantially deteriorate gait in freezers, an increase in cognitive demand is not exclusively responsible for FOG, once visual cues were able to overcome any interference evoked by the dual task [15].

Spildooren et al. [13] elucidated in their study of 2010 that a 360-degree turn in combination with a cognitive dual task is the most important trigger to cause FOG. Seven of fourteen participants froze during their protocol, but the number of FOG episodes is not presented in the results section.

In the work of Schaafsma [12] on the assessment of the effect of dopaminergic medication on distinct FOG subtypes at OFF state, nineteen participants were videotaped whilst walking 130 m during OFF and ON medication states. Three different observers characterized the type, duration, and clinical manifestations of FOG, and quantified FOG by analyzing the videotapes. During the OFF state, FOG was elicited by turn (63%), start (23%), walk through a narrow doorway (12%) and reach destination (9%) [12].

Schaafsma et al. [12] analyzed FOG while not only considering the type of FOG (leg movement observed), but also the FOG subtypes related to the trigger in order to determine if the response indicates whether levodopa improves FOG. The gait task was videotaped, and a video analysis was undertaken by three observers. The number of FOG episodes in Schaafsma et al. [12] is undisclosed, with their results only showing the percentage of FOG occurrence according to the triggers used.

In the study of Bartels et al. [4], patients were asked to stand up from a chair, walk 20 m, make a 360-degree turn to the right and 540-degree to the left and walk the same route back, ending with a turn to sit back in the chair. At the 10 m mark, the participants were to walk between two chairs which create a narrow path of 50 cm width. For the second task, patients were to walk an additional 50 m passing through two doorways [4]. Some patients could not walk two laps during the OFF state and were excluded from the study [4], the final number of volunteers was not presented in the paper.

A simple method was presented by Popovic et al. [8] for triggering and detecting FOG episodes using a series of stride force profiles recorded with force sensitive resistors. Data from nine participants were collected, and 24 FOG episodes were considered. Patients were asked to stand up from their chair, walk toward the room, walk through a doorway, reach a 13 m marker, make a 180-degree turn to the left and walk the route back. Their findings show that FOG most often occurs during turns and gait initiations.

In the study of Popovic et al. [8], the periods of FOG episodes were studied from three sources; videotapes, ground reaction forces and acceleration. Similar to the present work, video recordings were used for method validation. The method proposed by Popovic [8] was indeed simple, but not as effective as the method proposed in the present study in relation to the number of FOG episodes triggered.

Handojoseno et al. [20] demonstrated the potential of the EEG features extraction to give insights into the pathophysiology of FOG in PD. It was found that both power spectral density and wavelet energy could potentially act as biomarkers during FOG episodes. The dimension of data collection—5 h and 30 min of TUG test and 404 FOG episodes ranging from 1 to 220 s—were labeled by two clinicians specializing in movement disorders [20]. Popovic et al. [8] and Handojoseno et al. [20] have highlighted the importance of further exploration regarding a reliable method to provide quantitative measurements in the assessment of FOG in PD.

A study from Wang et al. [27] applied an in-place movement experiment for PD patients to provoke FOG and acquired a multimodal physiological signal (EEG, Acc, EOG, ECG), over 700 FOG episodes were provoked from 15 subjects, and most of these were provoked by the rapid turn condition. The subjects were examined at the OFF-medication state. There were five sessions of three conditions: (i) stepping in-place (SIP); (ii) half turning at a self-selected speed for 2 min; and (iii) half turning at a rapid speed for 2 min. Each session lasted about six minutes, for a total of 30 minutes of data collection [27]. Common sense leads us to conclude that the execution of in-place half turns at a rapid speed for several minutes, even at a self-selected speed, can cause dizziness, vertigo, malaise, discomfort, and a high risk of falling. The experiment setup proposed by Wang et al. [27] is efficient at triggering FOG episodes, however it is not practical for clinical care and could not be applied in frail patients.

In this study, a literature review was conducted to bring together the various methods applied in previous studies to induce FOG episodes in freezers. The innovation is the development of a physical mobility task that is practical and reliable for clinical care; uses known strategies that provoke FOG and which can be performed in a controlled environment without the use of complex technologies such as force platforms and treadmills; and which respects the frailty and limitation of a patient with Parkinson’s disease. The task was tested and validated using wireless inertial sensors and a camera. The advantage of assessing physical mobility in freezers during clinical practice and of considering FOG analysis for adequate treatment and follow-up is derives from its impact on reducing and overcoming FOG episodes, preventing falls, and increasing the quality of life of individuals affected by this sign of Parkinson’s disease.

### 2.2. Creating the Physical Mobility Task to Trigger FOG

Table 1 shows that the dual task is used in eight of the 17 studies, so we created two physical mobility tasks, one with a simple motor task and the other with a dual task, in which the participant performed the same simple motor task while also performing a cognitive task.

#### 2.2.1. Motor Task

The motor task consists of routine actions such as sitting and standing from a chair, walking in a straight line, passing through a doorway, and turning right and left. Figure 2 depicts the physical mobility motor task (MT) used in this study to evaluate FOG. The subject goes up from the chair, walks 3 m, and passes through a 67.5 cm-wide opening. The individual then moves 1.30 m to contour two obstacles, forming a path in the shape of ∞.

The following steps are carried out to complete the obstacles path:Initially, the subject performs a 360-degree turn to avoid the obstacle located on the same side as the subject’s most affected body part, as determined during the clinical evaluation by the physical therapist;The volunteer then moves toward the second obstacle and performs a complete 360-degree turn to avoid it;Finally, they move in the direction of the first obstacle and execute a second 360-degree turn to avoid it;Then they return to the direction of the 67.5 cm-wide opening.The task is complete once the subject is seated in the chair.

The selection of the opening width (67.5 cm) was based on a study conducted by Almeida et al. [30], who suggested a modified TUG test in which the volunteer walks through a door while performing the TUG test. The experiment was conducted with three different opening sizes: spacious (1.8 m), normal (0.9 m), and narrow (0.675 m), and their results demonstrate that the narrowest passageway had the greatest impact on the gait of the freezers.

#### 2.2.2. Cognitive-Motor Dual Task

The cognitive–motor dual task (DT) is executed in the same setting as the motor task. The patient performs the motor task described in Section 2.2.1 while performing the Digit Monitoring Task (DMT) [15], in which a random integer number (from 1 to 9) is assigned to each volunteer, and the researcher instructs the volunteer to silently count, without using fingers, the number of times the digit is announced over a loudspeaker. The audio track (Figure 3) was identical for all participants, and was transcribed so that the researcher had access to the correct digit. This task was performed three times for every subject and medication state (ON or OFF), resulting in a total of six trials. For each trial, a digit from the set of 1 to 9 was drawn without replacement.

At the end of the experiment, the researcher asks the participant how many times he heard the drawn digit. The entire data collection is recorded for later verification of the results, to check the volunteers’ response and compare with the actual number of times the digit appeared in the audio. The audio track used during the dual task is available in Appendix A.

To prevent gait synchronization with the audio track, the interval between auditory stimuli ranged from 100 to 1000 ms [15]. The duration of the audio was sixty seconds, which is the average time required to complete the proposed physical mobility motor task. Participants were instructed to continue counting the digit even if they completed the motor task prior to the conclusion of the audio.

### 2.3. Testing the Physical Mobility Motor Task

The study was conducted according to the guidelines of the Declaration of Helsinki, and all protocols were approved by the Ethics Committee. Informed consent was obtained from all subjects involved in the study. The experiment was performed in a place designated for the clinical care of Parkinson’s disease patients. Before carrying out the proposed physical mobility tasks, the motor task and the cognitive–motor dual task, participants performed the TUG test, a physical mobility task for gait analysis in Parkinson’s disease.

#### 2.3.1. Subjects

We included ten volunteers with PD [5,8] between the ages of 50 and 73 years old (mean: 60.6 years old; standard deviation: 7.29 years old). Table 2 displays the clinical characteristics of the volunteers, six of whom were female and four of whom were male. Table 2 displays the time of diagnosis in years, the duration of the OFF-medication state in hours, the MDS-UPDRS score during the OFF and ON states of medication, the Mini Mental Status Examination (MMSE) score, the New FOG Questionnaire (NFOG-Q) score, and the TUG score.

The volunteers were able to walk independently for 10 m and reported experiencing FOG in the prior month. FOG severity was rated using the New Freezing of Gait Questionnaire (new FOG-Q) [1,2,5,27,31,32]. All the participants had a FOG history with different severity and frequency [20].

The New FOG-Q consists of three sections. In Part I, the question “Have you experienced freezing episodes in the past month?” distinguishes between individuals with and without FOG. In Part II, five items with scores ranging from 0 to 4 assess the severity of FOG, and in Part III, three questions with scores ranging from 0 to 3 assess the impact of FOG on daily life activities.

Patients were excluded if there was evidence of cognitive impairments, which indicates a Mini Mental Status Examination (MMSE) less than nine [4,5,33].

All subjects were assessed first in the morning in the OFF-medication state, which is at least 12 h since the last intake of dopaminergic medication, then after data collection, they took their first dose of dopaminergic medication of the day and the experiment was repeated after 40–50 min, at the ON-medication stage [4,13,27,32]. Clinical data to rate severity of PD, as the MDS-UPDRS and the Hoehn and Yahr Scale, were collected during the OFF and ON states [2,4,32].

The subjects wore their regular footwear and no walking aid (cane or walker) [18,19]. A physiotherapist followed the subject during data collection to monitor possible loss of balance and prevent falls, but no physical assistance was given [19,32].

#### 2.3.2. Data Analysis

To detect FOG and characterize its occurrence we considered the definition of Bartels [4] in which an FOG can appear as a hesitation in the beginning of the movement that lasts more than one second; as a significative hesitation in the locomotion without an apparent reason; or when it looks like the volunteer fails when they try to begin or continue their movement.

The Movement Disorders Monitoring System (NetMD) [34,35], developed by the Spanish research group CAR-CSIC to analyze and to monitor movement disturbances remotely and continuously through inertial signals, was used to collect data from gait while participants performed the proposed tasks and from the consequent FOG episodes.

The locomotor band, when the volunteer walks normally, has frequency components ranging from 0.5 to 3 Hz, and the freeze band, when a FOG event is happening, has frequency components ranging from 3 to 8 Hz [6,11,36,37].

NetMD (Figure 4) is based on the combined action of an Android mobile phone with three smartwatches devices (Smartwatch3 SWR50 model, from Sony), with communication established via Bluetooth. Through this system, it is possible to acquire internal signals from the accelerometers and gyroscopes coupled in the smartwatches with a frequency of sampling rate of 50 Hz (temporal resolution of 20 ms). The system generates a text file with 10 columns that contain the values of the inertial signals for each smartwatch (time in milliseconds, sensor identification, battery status, accelerometer (m/s²) and gyroscope (rad/s) information on the x, y and z axes). The resulting file is stored in an Android mobile phone [35].

The system allows the copying of these text files to the computer, so that it is possible to process them in RStudio [34]. To increase the signal resolution, the collected signals were interpolated using splines, increasing the sampling frequency to 100 Hz.

A smartphone camera was used as an environment sensor. Several studies filmed the data collection because videotaping is a valuable tool for assessing movement [38]. Therefore, the environment in which the task to trigger FOG took place had a positioned camera to capture all the pathways. The technology available in smartphones allows high quality video recordings. Therefore, the data collection was filmed with a cellphone camera so that the physical mobility tasks could be watched by researchers to discriminate gait events and FOG episodes [12,19].

Three smartwatches with wireless inertial sensors were used, two of which were attached to the pelvis, over the two ends of the iliac spine [39] and one on the leg, over the fibula [6,11,40]. The sensors were attached to two adjustable belts, one on the hip and the other on the calf.

The sensor on the calf was placed on the side most affected by PD. This side was identified during the MDS-UPDRS Part III, the motor examination, executed by a qualified and experienced health professional.

The detection of FOG was carried out by an experienced researcher. Figure 5 shows a typical example of a recorded signal. The FOG episode is shadowed in gray. Figure 5A is the z axis of the accelerometer and Figure 5B is the y axis of the gyroscope, both from the sensor placed on the calf most affected by PD.

As shown in Figure 5, FOG can be detected using only the analysis of the accelerometer signal [26]. When a FOG episode occurs, the frequency of the accelerometer increases and the amplitude of the gyroscope decreases [23]. In addition to the signals of the three inertial sensors attached to the body of the volunteer, all FOG episodes were manually annotated by evaluating the video recordings [9] in conjunction with the inertial signals, using ATLAS [41].

## 3. Results

The results are presented to validate the proposed physical mobility tasks. Table 3 shows the number of FOG episodes during the TUG test, Table 4 shows the number of FOG episodes during the proposed physical mobility motor task (MT) and Table 5 shows the number of FOG episodes during the dual task (DT). Three trials were conducted in each medication state (ON or OFF) to increase the number of observations, thereby contributing to a more reliable result. The answers of the volunteers for the DMT during the dual task are presented in Appendix A.

Table 6 shows the information about the NFOG-Q score and the total number of FOG episodes for each volunteer during OFF and ON medication states and while performing the proposed physical mobility tasks, the motor task, and the cognitive–motor dual task.

Table 6 displays the total duration time of FOG episodes and the TF (as employed in several studies [9,15,23,25,27]). The latter represents the sum, in seconds, of the time difference between the beginning and end of each FOG episode. The results depict the number of FOG occurrences and total duration of FOG episodes for each participant during the OFF and ON medication states.

Table 7 compares the results of the present study with the results of the studies included in the literature review that could trigger and detect FOG episodes during data collection. Table 7 shows the author and year of the paper (see also Table 1), the sample size N, which is the number of participants who took part in the study, the number of freezers, which represents the number of participants who froze during the experiment, the number of FOG events, the recording time, and the number of trials.

Not all papers disclosed the duration of the trials, so recording time was not included in lines 1, 2, 3, and 5 of Table 7. The authors conclude that, analogous to what happened in our work, not all trials have a specific time to be completed. As a result the authors chose to analyze the number of trials indicated in the method sections of the studies—except Handojoseno et al. [20]—and compare this with the number of FOG events that occurred during the proposed physical mobility tasks—the motor task and dual task—during the OFF-medication state.

The three types of FOG according to the literature [8,10,11,12] are presented in Figure 6, Figure 7 and Figure 8. The figures show data from accelerometer and gyroscope on x, y and z axes placed on the right iliac spine. The waveforms shown in Figure 6, Figure 7 and Figure 8 are typical signals that occur during akinesia, shuffling and trembling in place.

Figure 6 shows one type of freezing—akinesia—of two volunteers, one for each row. This FOG type can manifest as a complete akinesia, in which the entire body is frozen, or partial akinesia, when only the lower body is frozen. Figure 7 shows the second type of freezing—festination or shuffling—of two volunteers, one for each row. This FOG type is characterized by a change in the normal gait rhythm. Finally, Figure 8 shows the third type of freezing—trembling in place—of two volunteers, one for each row. This FOG type happens when the lower extremities show signs of shaking while glued to the ground.

Table 8, based on the work of Podsiadlo and Richardson [18], illustrates how the proposed physical mobility motor task can be used to record changes in the duration and number of FOG events over time. A pair of specific improvements for freezers—a reduction of the time in seconds of FOG and the number of FOG episodes—is of particular interest.

## 4. Discussion

The results of this study support our hypothesis that it is possible to cause FOG in a controlled environment with a short and simple physical mobility motor task, using limited space, without the use of complex technologies as force platforms and treadmills, and respecting the fragility and limitation of a Parkinson’s disease patient. Besides that, the findings show that it is possible to detect FOG and to distinguish FOG types, i.e., freezing, shuffling, and trembling in place, by means of inertial sensors placed on the hips and calf. In our population of Parkinson’s disease patients, the proposed physical mobility tasks to assess FOG, both motor task and dual task, were practical and reliable.

Time in seconds to complete TUG provides a score that is an objective mean of the consequent functional changes of an individual over time [18]. The TUG score for each volunteer who participated in this study is shown in Table 2. In the same sense, the proposed physical mobility motor task to assess FOG can be used either as a screening test or a descriptive tool, as shown in Table 8.

The New FOG-Q is reliable [32] and has been used in several experiments [1,2,5,27]. Therefore, it is considered an important measurement for the elucidation of FOG severity during the assessment of freezers in Parkinson’s disease. Table 2 shows that the highest score of TUG and of the New FOG-Q belongs to volunteer 9, a subject that experienced several FOG episodes during OFF and ON medication states. Volunteer 8 has the second highest score for New FOG-Q, with 14 FOG events, and nine minutes and three seconds spent in FOG condition during data collection (See Table 6). Volunteer 5 has the third highest score of New FOG-Q and froze during OFF and ON medication states. Only volunteers 5 and 9 froze under both medication conditions. Volunteer 2 had a significantly higher New FOG-Q score; however, the subject froze only three times.

Volunteers 3 and 4 have the highest number of FOG events, both have a New FOG-Q score greater than 19. For the volunteers with New FOG-Q scores lower than 14 (volunteers 1 and 10), the number of FOG events was 1 or 0 and the time spent frozen was less than 2 s. Only one volunteer did not freeze during the experiment.

It is not possible to state with certainty the reasons why volunteers 3 and 4 had the highest number of FOG, since this symptom is still a poorly understood phenomenon, and its pathophysiology, according to the literature, is still unclear according. Both volunteers do not stand out from the others in relation to age, time of PD diagnosis, the TUG score, nor the total MDS-UPDRS score, however when considering the New FOG-Q, both have a score greater than 19. Even so, the highest scores for the New FOG-Q belong to volunteers 9, 8, 5 and 2.

Motor performance among PD patients generally shows large variability. This was also the case among the group of patients who participated in this study. For example, during nonfreezing episodes, some patients maintained a regular gait that could hardly be distinguished from that of healthy elderly people, while others had slow and unstable gait [37].

Bachlin [37] stated that a limitation in FOG studies is that the controlled environment and the presence of a physiotherapist may reduce the likelihood of FOG in patients that do not experience any FOG events during data collection and where researchers are not able to explain why two volunteers did not have any FOG during their study.

The video recordings of our data collection show that volunteer 1, the one who did not freeze during our data collection, used strategies for overcoming FOG episodes, such as adapting a faster-than-normal rhythm and lifting the leg higher than usual to walk [5].

Table 6 shows that the time spent frozen is not directly correlated with the number of FOG events. For example, volunteer 3 had 39 FOG events during their OFF-medication state while performing the motor task and the dual task, but the time they spent frozen was less than 3 min, while volunteer 8 had 14 FOG events performing the same tasks, but they spent more than 9 min in a FOG condition while performing the proposed tasks. The same happens with volunteer 4, who had 21 FOG events and spent more than 11 min in a FOG condition. One can conclude that volunteer 3 has a high number of FOG events, but is able to de-FOG fast, which means that they are able to get out of the FOG condition in less time than volunteers 4 and 8, for example. It can be concluded that, due to the variability of the signs in Parkinson’s disease and the different progressions of the disease in each individual, volunteers 3 and 4 were more sensitive to the triggers used in the proposed tasks.

Velik et al. [27] and Alvarez et al. [33] recorded daily life motion to detect specific motor patterns of the limbs, since it is often difficult to observe these during clinical visits [2]. However, to achieve this feat, expensive technology, large available space, and a previously trained staff are required to operate the cameras and sensors. Additionally, there is a need for storage capacity so that all recordings are saved and can be later watched by experienced evaluators for the detection of FOG. Meanwhile, the present work raises the possibility that FOG assessment can be carried out, supported by health professionals, in a limited space by performing a physical mobility task designed to trigger FOG events. Furthermore, the detection of FOG events for posterior analysis can be executed with the use of inertial sensors and video recordings from a cell phone camera.

The proposed method to assess FOG, when combined with a mobile phone and three smartwatches equipped with inertial sensors, allows one to trigger and to detect FOG episodes in a controlled environment using a physical mobility task. We were able to create an accurate representation of daily life situations to cause FOG by developing a task that includes all the movements highlighted in the literature as potential triggers to FOG episodes, for instance, to initiating gait, walking through a narrow doorway, making left and right 360-degree turns, and reaching a destination.

It was possible to detect FOG using inertial sensors placed on the hips and calf of the subjects while they performed the proposed physical mobility tasks (MT and DT). Table 6 shows the number of FOG episodes and the total time in seconds of FOG duration. Table 4 and Table 5 show the total number of FOG episodes for each trial of the motor task and the dual task per volunteer. This number is considered high when compared with the work of Popovic et al. [8], Beck et al. [15] and Cando et al. [23] presented in Table 7, and when compared with the number of FOG events caused by the TUG test, displayed in Table 3.

Dopaminergic medication has a significant effect on the occurrence of FOG [4,12]. Clinical experience suggests that most patients who experience FOG improve with the dopaminergic medication, however FOG persists in a milder form [4]. The results of Schaafsma [12] suggest the medication increases the threshold of FOG occurrence but does not cure the symptoms. Table 3, Table 4, Table 5 and Table 6 show the difference in number of FOG comparing OFF and ON medication states while performing the proposed motor tasks. They show that 125 FOG episodes occurred during the OFF state and 19 FOG episodes occurred during the ON state. Therefore, the proposed method is able to detect the reduction in FOG events when patients are using dopamine.

The comparison between volunteers young and older than 60 showed that, for volunteers younger 60 years old, the total number of FOG was 64, distributed in 32 FOG events for the motor task and 32 for the dual task. TUG did not trigger any FOG episode during OFF medication state for this subgroup. During the ON medication state, there were no FOG episodes either.

For volunteers older than 60, the total number of FOG was 80, distributed in such a way that there were 18 for TUG, 21 for the motor task and 22 for the dual task during the OFF medication state. During the ON medication state, there were six FOG episodes in TUG, seven FOG episodes in the motor task and six FOG episodes in the dual task.

An important aspect to note of Table 4 and Table 5 is the number of FOG events triggered by the motor task (MT) and the dual task (DT). The simple motor task caused 53 FOG episodes while the dual task caused 54 during the OFF medication state. The TUG test, which is widely used in the clinical evaluation of PD patients, triggered 18 FOG episodes during the OFF medication state. During the ON medication state, there were seven FOG episodes for MT, six for DT and six for the TUG test. The close results between the proposed tasks may indicate that the dual task is not crucial to trigger FOG, as at one point both the simple motor task and the dual task caused 60 FOG episodes each. Furthermore, the results show how effective novel physical mobility tasks are in triggering FOG to assess freezers in Parkinson’s disease, as they were able to cause three times more FOG episodes during the OFF medication state than the TUG test.

Studies that have investigated dual task [13,15,21,22,29], have demonstrated that cognitive load has a negative effect on the gait of patients with PD, however, when it comes to FOG-provoking strategies, the dual task is non-essential according to our findings. Considering the equipment and resources needed to complete the dual task, e.g., a device to play the audio track with the numbers for the Digital Monitoring Task, a speaker, a number draw, and the complexity of the dual task itself—not only for the person who is performing the dual task but also to the researcher group—one can conclude that, when the physical mobility motor task is specially designed to cause FOG episodes using triggers of movement and environmental constraints, that the dual task is not fundamental.

Figure 6, Figure 7 and Figure 8 show, respectively, the distinct FOG types, freezing, festination, and trembling in place, defined by Thompson and Marseden [8]. In the work of Schaafsma [12], three different observers characterized the type, duration, and clinical manifestations of FOG, and quantified it by analyzing their videotapes. In the study of Bartels et al. [4], three observers independently watched videotapes of the tasks and elucidated the number of FOG episodes. A FOG episode was considered to take place if a patient hesitated for one second or more [4]. The researchers clarified that by exclusively using the video rating method, they might have missed very brief FOG episodes [4]. The analysis of FOG episodes using only video recordings represents a subjective assessment of the patient, because it depends on the observer’s experience and expertise, the shooting angle, and the quality of the video. It has been proven by several studies [8,20,27] that the use of sensors optimizes data collection.

The literature review showed that 11 studies were able to trigger FOG in a controlled environment. Table 7 shows the papers over time that disclose the number of FOG episodes during the experiment. Table 7 displays the total number of volunteers included in the research; the column titled ‘Number of Freezers’ shows the actual number of volunteers who froze during data collection [15,23,27]. The difficulty of accessing the FOG and causing it in a controlled environment is also disclosed in the studies of Saad et al. [6] and Jovanov et al. [6,9] in which, to test a new equipment, the researchers had to simulate FOG episodes themselves.

Table 8 is an example of how the proposed physical mobility motor task can be used to record changes in the duration and number of FOG events over time and, per Figure 2, in a setting where freezers perform regular tasks, thus allowing the examiner to assess the FOG symptom during a clinical visit.

An intervention designed by Rawson [5] to reduce FOG in PD was tested in seven patients. The participants completed what the authors called a ‘FOG boot Camp’, a six-week program with one and a half hour classes each week designed and taught by two specialists in neurological and geriatric physiotherapy, respectively [5]. The classes had education and group discussions on strategies for overcoming a FOG episode. This was followed by practicing these strategies in environments designed to trigger FOG, such as sharp turns, narrow pathways, doorways, and turns to sit in a chair [5].

The applied strategies were based on existing literature and included sensorial cues (auditory, visual, and vibratory) and self-initiated strategies that required executive functioning and attentional processes and included activities such as lifting one leg higher than usual, walking sideways, moving one foot backward before walking, making wider turns, shifting leg weight, and imagining a clock on the ground to help with turning [3,5]. The main objective of Rawson et al. [5] was to determine the feasibility, safety, and acceptability of a once-weekly community-based group intervention. Participants had favorable feedback and showed reduced FOG.

External rhythmic cues have been found to be effective in overcoming FOG [19,23,42]. However, the continuous presence of an auditory, visual, or vibratory cue may reduce effectiveness and disturb normal social activity [27]. Therefore, taking the dopaminergic medication correctly and learning self-initiated strategies to reduce or overcome FOG episodes is the best current alternative for freezers.

The proposed physical mobility motor task could be used as a tool to measure functional changes, to record changes in the duration and number of FOG events over time, to assess the impact on gait of a medication dosage change, and to test the acknowledgement and effectiveness of learned self-strategies to overcome FOG. The improvement of gait mobility and the decrease frequency of FOG events are the goals when assessing gait in freezers [5].

Further work, with an increased number of volunteers, is required to understand the influence of age, the time of diagnosis and the New FOG-Q score on results and to improve the analysis. Additionally, the next stage of the research should be a feature extraction from the inertial signals, to better understand the signal dynamics and how the accelerometer and gyroscope signals from the three sensors placed in the volunteer’s body behave during an FOG event.

## 5. Conclusions

To design the proposed physical mobility tasks, we carried out a literature review and determined that it was possible to set up a practical motor task to assess freezers in Parkinson’s disease. This paper presents the development of a simple physical mobility task capable of causing FOG in a controlled environment, using known triggers. A group of ten volunteers who experience FOG in daily life participated in the validation of the proposed method, which was carried out using inertial sensors and video recordings. Accelerometers and gyroscopes were able not only to detect FOG episodes but also to show the different types of FOG that depend on leg movement (akinesia, shuffling and trembling). The proposed tasks caused 120 FOG episodes. Volunteers froze more during the OFF medication state compared with the ON state and the number of FOG episodes was higher while performing the proposed physical mobility tasks than the TUG test. The proposed method may be used to complement clinical examination in the assessment of freezers.

## Figures and Tables

**Figure 1 healthcare-11-00409-f001:**
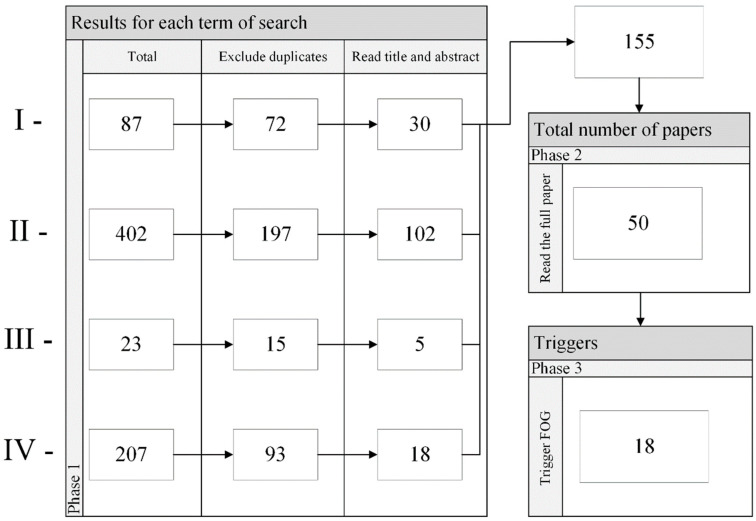
Flow diagram depicting the strategy adopted for literature review. The search terms are identified by I, II, III and IV.

**Figure 2 healthcare-11-00409-f002:**
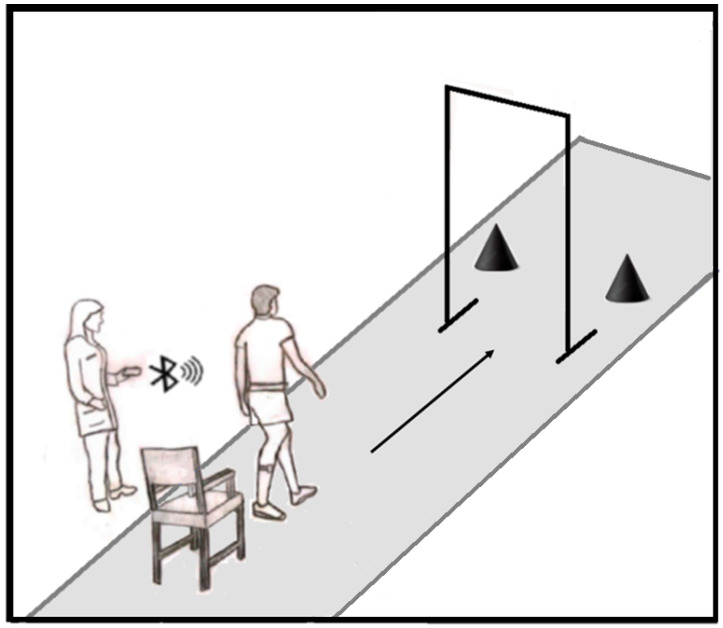
Representation of the simple physical motor task that causes FOG.

**Figure 3 healthcare-11-00409-f003:**
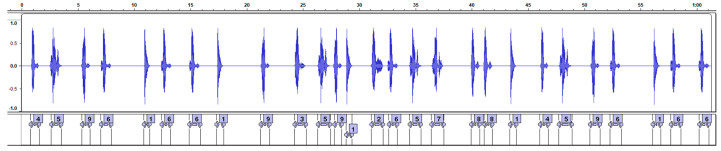
Screenshot of Audacity. The audio track is shown (top) together with the digits (bottom). This audio track was used for all dual-task trials. The duration of the track is sixty seconds.

**Figure 4 healthcare-11-00409-f004:**
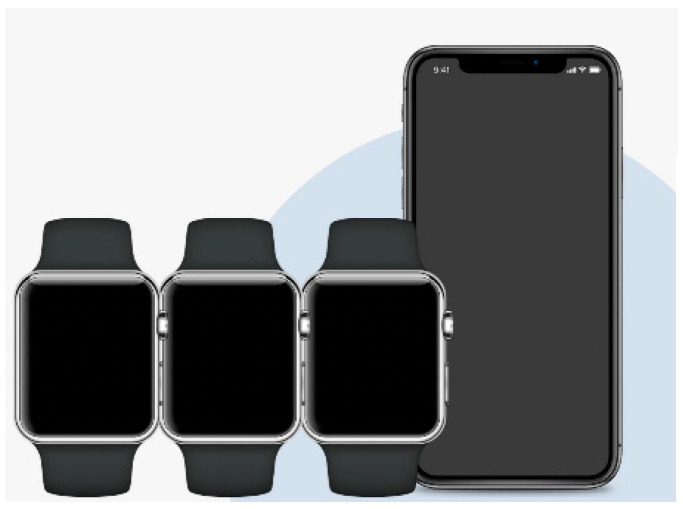
Representation of the NetMD system, with a mobile phone with three smartwatches devices.

**Figure 5 healthcare-11-00409-f005:**
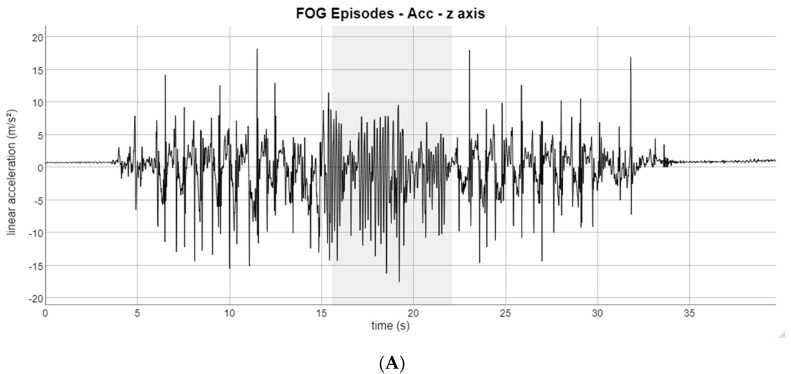
Example of a FOG episode shadowed in gray detected by the sensor placed on the volunteer’s most affected calf. Subfigures (**A**,**B**) represent the waveform of the inertial signal during the TUG test (**A**) Linear acceleration in the z axis of the accelerometer (**B**) Angular velocity in the y axis of the gyroscope.

**Figure 6 healthcare-11-00409-f006:**
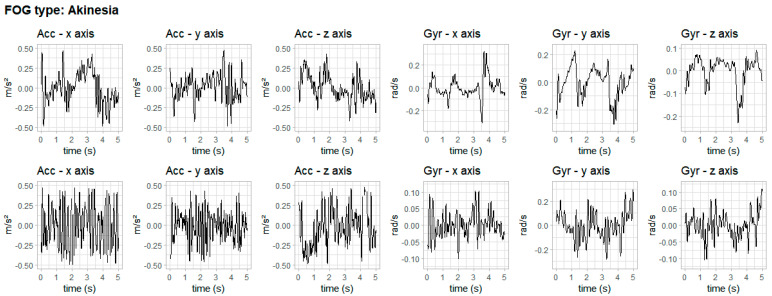
Typical waveforms of inertial signals during akinesia. Row 1 shows akinesia in one volunteer and row 2 shows akinesia in another volunteer.

**Figure 7 healthcare-11-00409-f007:**
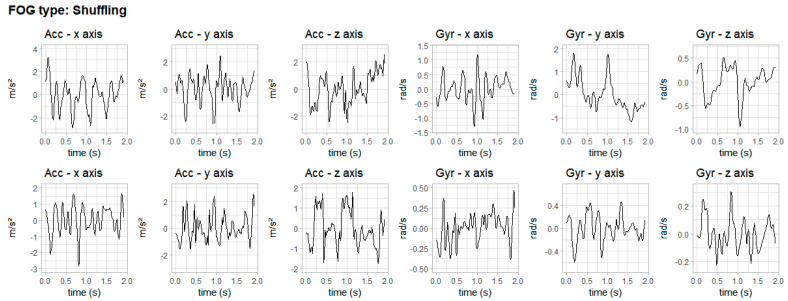
Typical waveforms of inertial signals during festination. Row 1 shows festination in one volunteer and row 2 shows festination in another volunteer.

**Figure 8 healthcare-11-00409-f008:**
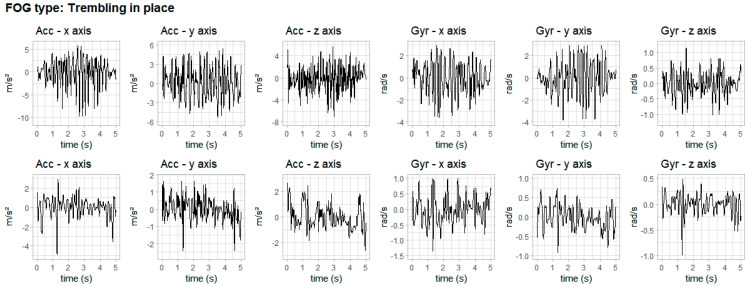
Typical waveforms of inertial signals during trembling in place. Row 1 shows trembling in place in one volunteer and row 2 shows trembling in place in another volunteer.

**Table 1 healthcare-11-00409-t001:** Papers included in the literature review with triggers that are able to cause FOG episodes.

Author and Year of Publication	Sit and Stand from a Chair	Narrow Doorway	Reach a Destination	180-Degree Turn	360-Degree Turn	Dual Task
SCHAAFSMA (2003) [12]	X	X	X	X	X	
BARTELS (2003) [4]	X	X			X	
JOVANOV (2009) [9]	X			X		
SPILDOOREN (2010) [13]				X	X	X
POPOVIC (2010) [8]	X	X		X		
VELIK (2012) [19]	X	X	X	X	X	X
SHINE (2013) [14]						X
BECK (2015) [15]		X				X
HANDOJOSENO (2015) [20]	X			X		
KILLANE (2015) [21]		X				X
WAECHTER (2015) [22]		X				X
CANDO (2016) [23]	X		X	X	X	
TARD (2016) [24]						X
BERTOLI (2017) [1]				X	X	
MURALIDHARAN (2017) [25]		X				X
ALVAREZ (2018) [26]	X	X	X	X	X	
WANG (2020) [27]				X		
PRESENT STUDY	X	X	X	X	X	X

**Table 2 healthcare-11-00409-t002:** Clinical characteristics of volunteers.

V	Sex	Age (Years)	Time of PD Diagnosis (Years)	New FOG-Q	MMSE	Time OFF	Part II MDS-UPDRS	Item 2.13 of the MDS-UPDRS (Gait Freezing)	Part III MDS-UPDRS		Total MDS-UPDRS Score		TUG Score	
									OFF	ON	OFF	ON	OFF	ON
1	M	50	15	14	27	12 h	18	2	62	50	80	68	2	2
2	F	51	7	23	28	13 h	16	2	49	39	65	55	2	2
3	M	57	15	19	15	13 h	21	1	51	26	72	47	2	1.33
4	F	59	12	21	27	13 h	10	1	20	7	30	17	2	2
5	F	63	10	23	26	13 h	12	1	53	32	65	44	3.33	2
6	F	65	14	16	28	10 h	6	1	35	37	41	43	2	2
7	F	66	25	15	24	13 h	22	2	94	57	116	79	3.66	2
8	M	68	6	23	28	13 h	9	1	49	37	58	46	2.66	1
9	M	73	6	26	27	12 h 30	44	3	115	100	159	144	4	4
10	F	56	15	12	25	13 h 30	13	1	47	32	60	45	2	2

**Table 3 healthcare-11-00409-t003:** Number of FOG episodes for each volunteer and the total number of FOG episodes during the OFF- and ON-medication states for the TUG test.

v	TUG							
	OFF				ON			
	TUG1	TUG2	TUG3	TOTAL	TUG1	TUG2	TUG3	TOTAL
1	0	0	0	0	0	0	0	0
2	0	0	0	0	0	0	0	0
3	0	0	0	0	0	0	0	0
4	0	0	0	0	0	0	0	0
5	2	1	0	3	0	0	1	1
6	0	0	0	0	0	0	0	0
7	1	0	0	1	0	0	0	0
8	0	3	4	7	0	0	0	0
9	2	3	2	7	1	2	2	5
10	0	0	0	0	0	0	0	0
Total number of FOG—TUG OFF				18	Total number of FOG—TUG ON			6

**Table 4 healthcare-11-00409-t004:** Number of FOG episodes for each volunteer and the total number of FOG episodes during the OFF- and ON-medication states for the physical mobility motor task (MT).

v	MT							
	OFF				ON			
	MT1	MT2	MT3	TOTAL	MT1	MT2	MT3	TOTAL
1	0	0	0	0	0	0	0	0
2	0	0	0	0	0	0	0	0
3	7	9	7	23	0	0	0	0
4	2	3	3	8	0	0	0	0
5	2	3	1	6	1	0	0	1
6	1	0	0	1	0	0	0	0
7	1	2	0	3	0	0	0	0
8	2	1	1	4	0	0	0	0
9	1	5	1	7	2	2	2	6
10	0	0	1	1	0	0	0	0
Total number of FOG—MT OFF				53	Total number of FOG—MT ON			7

**Table 5 healthcare-11-00409-t005:** Number of FOG episodes for each volunteer and the total number of FOG episodes during the OFF- and ON-medication states for the dual task (DT).

v	DT							
	OFF				ON			
	DT1	DT2	DT3	TOTAL	DT1	DT2	DT3	TOTAL
1	0	0	0	0	0	0	0	0
2	2	0	1	3	0	0	0	0
3	3	6	7	16	0	0	0	0
4	5	5	3	13	0	0	0	0
5	2	1	0	3	0	0	2	2
6	1	0	0	1	0	0	0	0
7	0	2	0	2	0	0	0	0
8	4	4	2	10	0	0	0	0
9	1	2	3	6	1	2	1	4
10	0	0	0	0	0	0	0	0
Total number of FOG—DT OFF				54	Total number of FOG—DT ON			6

**Table 6 healthcare-11-00409-t006:** Information about the NFOG-Q score and the total number of FOG episodes considering the sum of events in the mobility motor task and the dual task.

V	NFOG-Q	OFF		ON	
		Number of FOG	TF (s)	Number of FOG	TF (s)
1	14	0	0	0	0
2	23	3	4.498	0	0
3	19	39	179.735	0	0
4	21	21	691.521	0	0
5	23	9	27.448	3	4.922
6	16	2	6.273	0	0
7	15	5	12.918	0	0
8	23	14	571.886	0	0
9	26	13	88.480	10	64.821
10	12	1	1.769	0	0
Total		107	1584.528	13	69.743

**Table 7 healthcare-11-00409-t007:** Number of FOG episodes triggered by each experiment during OFF medication state.

Author	N	Number of Freezers	Number of FOG Events	Recording Time	Number of Trials
Bartels 2003 [4]	19	---	237	---	57
Popovic 2010 [8]	9	---	24	---	18
Beck 2015 [15]	20	4	23	---	240
Handojoseno 2015 [20]	16	---	404	5 h 30 min	---
Cando 2016 [23]	5	4	11	---	10
Alvarez 2018 [26]	18	---	200	8 h 20 min	700
Wang 2020 [27]	15	15	700	22 h 30 min	225
**Present study**	**10**	**9**	**107**	**---**	**60**

**Table 8 healthcare-11-00409-t008:** An example of how the proposed physical mobility motor task can be used to record changes in the duration and number of FOG events over time.

Patient	Age	Date	Walking Aid	New FOG-Q	State	TFOG	Number of FOG Events
1	57	1 March	---	12	OFF	6 s	2
					ON	0 s	0
		1 May	---	12	OFF	30 s	6
					ON	0 s	0
		1 July	---	13	OFF	8 s	3
					ON	0 s	0
2	63	10 April	cane	19	OFF	27 s	4
					ON	2 s	1
		10 July	cane	20	OFF	32 s	6
					ON	0 s	0
		10 October	cane	20	OFF	30 s	8
					ON	0 s	0

## Data Availability

The datasets generated in the current study are not publicly available due to the ethical restrictions preventing public sharing of data. A non-identified set may be requested after approval from the Review Board of the Institution. Requests for the data may be sent to the corresponding author.

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
