# Peer review of "A Novel Physical Mobility Task to Assess Freezers in Parkinson’s Disease"

_healthcare, 2023, doi:10.3390/healthcare11030409_

Round 1

Reviewer 1 Report

The work is related to the detection of the freezing of gait in Parkinson's disease exploiting a new mobility task to induce the FOG episodes in patients in controlled environments. The work is interesting and relevant from the research point of view.

However, the literature review is quite long and it is not clear where the description of the novelty of the work starts. Maybe a new subparagraph for the description of the novelty is needed.

The authors use on the shelf devices for the data acquisition. Also if it is of secondary importance with respect to the main purpouse of the work, it would be better to explain how the data are acquired, the sampling frequency of the acquired vibration signals, the bandwidth of the sensors used and characteristics in terms of signal dynamic and frequency components of the acquired vibration signals. Those informations will be very usefull for example to perform new researchs devoted to the realization of ad-hoc sensor nodes designed for the detection of FOG episodes.

Other minor issues:

The sentence in line 331 is incomplete.

The Y scales in figures 6,7,8 must be adapted to the signal dynamics. Moreover it is not clear why there are duplicated graphs for the three axis of the accelerometer and the three axis of the gyroscope for each FOG type. In the description is not clear what they represents.

Reviewer 2 Report

In this manuscript, authors have presented a simple physical mobility task capable of causing FOG in a controlled environment, using known triggers.

Overall, manuscript is well written especially the introduction and literature review (the key to this manuscript). However, the conducted experiment can be well improved such as increasing volunteers etc. 

Particularly, in the scope of this manuscript, I would like to accept it with few revisions. 

1.  Line 98 "The main databases used in our review were IEEE, 98 Pubmed, Lilacs, and Medline." Please mention that these are enlisted in Table 1. 

2. The motor task section 2.2.1, second paragraph starting from line 336. This paragraph should be in points to make it more clear.

3. The reasoning have been a little on the lower side. The analysis can be well improved. I would like to mention paragraph starting from line 453. Why the volunteers 3 and 4 had the highest FOG events? What is your analysis that one volunteer didn't freeze. Is it related to age difference as you have range from 50 to 73. Did you see the results close to 70? or there is a big difference? 

4. Similarly, in above mentioned paragraph (Starting from line 453) why one of the volunteer didn't froze? 

5. Since you have volunteers in age range of 50 to 73, I would like to see the comparison between pre and post 60. However, the 10 volunteers may not be enough for this, so I would recommend to make it possible in future. 

Round 2

Reviewer 1 Report

I would like to thank the authors for their responses to the reviews. The work is now very complete also from the point of view of the measurements. I suggest accepting it in its current form.